# Regional supervised learning of inhibitory control strength from cortical sulci

**Joël Chavas**                                              JOEL.CHAVAS@CEA.FR
**Aymeric Gaudin**                                          AYMERIC.GAUDIN@CEA.FR
**Denis Rivière**                                          DENIS.RIVIERE@CEA.FR
**Jean-François Mangin**                              JEAN-FRANCOIS.MANGIN@CEA.FR
*NeuroSpin, CEA Saclay, Université Paris-Saclay, France*

**Editors:** Accepted for publication at MIDL 2024

## Abstract

The human cortical brain is folded and is highly variable among individuals. We ultimately want to quantify how cortical folding relates to clinically relevant parameters. Here, using supervised convolutional networks on the human connectome project (HCP) dataset, we learn to predict inhibitory control strength, using all the information from the cortical folds. As we want to focus on the shape of the folds—which are supposed to remain identical throughout adult life—, we don't use the full MRI images but the cortical skeletons, which are negative casts of the brain. As we expect putative folding patterns to be local, we scatter the supervised learning over 24 sulcal bilateral regions on the two hemispheres and apply an ensemble method to each region. We found the strongest significant correlations between inhibitory control and cortical sulci in the frontal marginal, the central sulcal, and the cingulate regions.

**Keywords:** deep learning, cortical sulci, folding patterns, inhibitory control

## 1. Introduction

The cortical brain is folded, with a high interindividual variability. Links between cortical folding and clinically relevant parameters are often related not to the presence or absence of identified sulci (Ono et al., 1990; Borne et al., 2020) but to their particular shape or folding pattern. The motivation to study and characterize folding variability is twofold. First, as folds form early in life, folding patterns open a window into the early process of neurodevelopment. Second, specific folding patterns can be used as biomarkers of what happens during neurodevelopment, like markers of inhibitory control in juveniles (Cachia et al., 2016).

Here, we build a supervised benchmark on a neuropsychological parameter: inhibitory control, measured with the Flanker test (Eriksen and Eriksen, 1974). As cortical folding patterns are likely local geometrical entities, newly defined benchmarks should be learned on a regional scale, not on the whole brain. Thus, we run supervised deep learning algorithms and an ensemble method for the Flanker test measures prediction on 24 bilateral sulcal regions (Fig. 1).

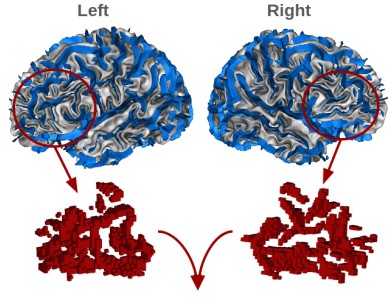

Figure 1: A bilateral regional supervised deep learning algorithm. Inputs are bilateral regional crops (in red) of the complete cortical skeleton (in blue).

## 2. Methods and Experiments

**Datasets and stratification.**  To access the Flanker test measurement—which measures both a participant's attention and inhibitory control—, we use the human connectome project dataset (Van Essen et al., 2013). We keep subjects on the third lower range and the third higher range of Flanker test measurements adjusted with age (Flanker_AgeAdj), categorized respectively as low-flanker and high-flanker (746 subjects total). We stratify train, validation, and test sets according to flanker category, gender, and zygosity [1].

**Description of the inputs.**  All subjects' brain magnetic resonance images (MRIs) were processed with the Morphologist pipeline from the BrainVISA software [2] to get their cortical sulcal skeleton—a negative cast of the brain folds—. They are then affinely normalized to a standard brain referential (ICBMc2009) and resampled with a 2 mm cubic voxel size.

**Sulcal region selection and masking.**  Borne et al. (2020) gives a nomenclature of 62 sulci per hemisphere. We define 24 overlapping sulcal regions in the standard ICBM referential from these sulci (Fig. 2A). In short, using a database manually labeled for this nomenclature of sulci, we construct a localization map for each sulcus, in which each voxel contains the number of subjects with a voxel of that sulcus at that position (Chavas et al., 2022). To define the sulcal region crop, we take the union (intersection in the case of the cingulate region) of the binarized maps of sulci included in the definition of the sulcal region and dilate it by 5 mm. Using these masks, defined only on the manually labeled dataset, we finally compute $2 \times 24$ crops (24 crops per hemisphere) for each subject [3] on the HCP dataset.

**Hyperparameter search and model evaluation.**  After an initial structural search starting from the backbone defined in Gaudin et al. (2024) [4], 45 models are run for each couple of regions with a ReLU projection head activation over a learning rate on $[10^{-4}, 10^{-2}]$ (log-uniform), over the drop rate on $[0.05, 0.3]$ and a gamma scheduler with step 10 and gamma on $[0.7, 1.0]$, whereas the maximum rotation angle applied randomly to input images is set to 6. The models are trained for 100 epochs and evaluated using the area under curve (AUC) of the receiver operating characteristic curve.

Flanker classification results are highly dependent on random initialization, as we observe either training underfit, training overfit, or convergence with the same set of hyperparameters. Thus, from the 45 models, only the five models having the smallest $criterion = 1 - AUC_{val} + \frac{1}{2}max(AUC_{train} - AUC_{val}, 0)$, which maximizes the validation AUC and penalizes the training overfit, are selected for the analysis.

To improve the classification, we turn to the ensemble averaging model. We report the AUC over the average softmax prediction for all selected models. We compute the ensemble standard deviation as the standard deviation of 100 bootstrap samples.

---

1. We also guarantee that monozygotic and dizygotic twins are always on the same set to avoid data leakage between sets of subjects with related genetic information

2. https://brainvisa.info

3. https://github.com/neurospin/deep_folding

4. The final CNN backbone is a double 6-layer convolutional network (each layer being a Conv3D-BatchNorm3D-DropOut3D-LeakyReLU), followed by a [20,10,10,2] non-linear dense projection head. The code is available at https://github.com/neurospin-projects/2023_agaudi_jchavas_folding_supervised

# 3. Results

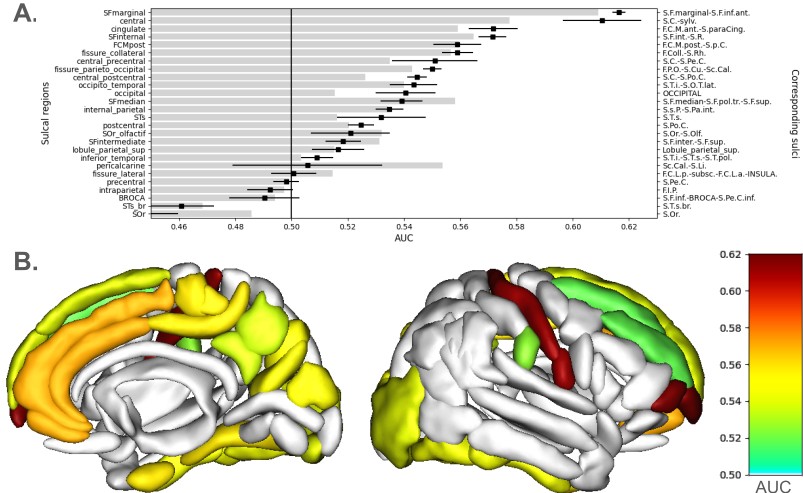

Figure 2: *Test AUC results for the Flanker test (inhibitory control measurement) per sulcal region.* **A)** AUC results of all five selected models per sulcal region. The bar plot is the AUC average. The scatterplot and the error bars represent the mean ensemble AUC and the ensemble bootstrap standard deviation. **B)** Ensemble AUC results mapped on the sulcus SPAM (Statistical Probabilistic Anatomy Map (Perrot et al., 2011)) model of the right hemisphere. When one sulcus belongs to two sulcal regions, we assign the smallest AUC value to this sulcus. We have grayed the regions when AUC=0.5 is within 2.7 bootstrap standard deviations of the ensemble AUC.

The ensemble mean AUC for the Flanker test measurement is higher for the frontal marginal region (AUC goes up to $0.62 \pm 0.002$), the central sulcal region, and the cingulate region (anterior and posterior)(Fig. 2). The frontal lobe also correlates significantly, except for the precentral and occipito-temporal regions. No correlation is detected in the Superior Temporal Sulcus, in the BROCA, the intraparietal, the precentral, the lateral fissure, the pericalcarine, and the inferior temporal regions.

These results align with the literature, which demonstrated a correlation between folding patterns in the cingulate region and inhibitory control (Cachia et al., 2016), as well as a correlation between inhibitory control and the damage studies in the intermediate and median frontal region (Aron et al., 2003; Floden and Stuss, 2006).

# 4. Conclusions and perspectives

Using supervised deep learning and ensemble methods for inhibitory control assessment, we found sulcal regions that predict inhibitory control strength significantly. Moreover, the strength of the prediction is not distributed equally among all regions, as some sulcal regions have no prediction power, showing that what is found is not a global feature as would be, for instance, a global change in sulcal depth. Thus, these results can be used as supervised AUC benchmarks of folding patterns for each bilateral couple of sulcal regions for Flanker test measure prediction.

## Acknowledgments

This works was funded by the grants ANR-19-CE45-0022-01IFOPASUBA and ANR-20-CHIA-0027-01FOLDDICO.

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
