# OpenReview forum: "Regional supervised learning of inhibitory control strength from cortical sulci"
_MIDL.io/2024/Short_Papers — MIDL 2024 Short Papers_

### Official Review · Reviewer_mM2A · 2024-04-24

**Confidence:** 5
**Final Rating:** 5

**Review:**

This paper presents supervised ensemble learning based on 24 sulcal bilateral region features in the brain of inhibitory control strength. Sulcal region analysis in the brain is a newer direction than volumetric ROI analysis, so it could attract interest from the MIDL community. It also shows that its findings are aligned with existing clinical research.

---

### Decision · Program_Chairs · 2024-04-26

Accept